

# Diversity and community structure of marine microbes around the Benham Rise underwater plateau, northeastern Philippines

Andrian P. Gajigan[1,2], Aletta T. Yñiguez[1], Cesar L. Villanoy[1], Maria Lourdes San Diego-McGlone[1], Gil S. Jacinto[1] and Cecilia Conaco[1]

[1] Marine Science Institute, University of the Philippines Diliman, Quezon City, Philippines
[2] Current affiliation: Department of Oceanography, University of Hawaii at Manoa, USA

## ABSTRACT

Microbes are central to the structuring and functioning of marine ecosystems. Given the remarkable diversity of the ocean microbiome, uncovering marine microbial taxa remains a fundamental challenge in microbial ecology. However, there has been little effort, thus far, to describe the diversity of marine microorganisms in the region of high marine biodiversity around the Philippines. Here, we present data on the taxonomic diversity of bacteria and archaea in Benham Rise, Philippines, Western Pacific Ocean, using 16S V4 rRNA gene sequencing. The major bacterial and archaeal phyla identified in the Benham Rise are Proteobacteria, Cyanobacteria, Actinobacteria, Bacteroidetes, Marinimicrobia, Thaumarchaeota and, Euryarchaeota. The upper mesopelagic layer exhibited greater microbial diversity and richness compared to surface waters. Vertical zonation of the microbial community is evident and may be attributed to physical stratification of the water column acting as a dispersal barrier. Canonical Correspondence Analysis (CCA) recapitulated previously known associations of taxa and physicochemical parameters in the environment, such as the association of oligotrophic clades with low nutrient surface water and deep water clades that have the capacity to oxidize ammonia or nitrite at the upper mesopelagic layer. These findings provide foundational information on the diversity of marine microbes in Philippine waters. Further studies are warranted to gain a more comprehensive picture of microbial diversity within the region.

## INTRODUCTION

Microbes play a critical role in marine ecosystem structure and ocean biogeochemistry. Thus, it remains essential to understand the vast diversity of microbes living in our seas. Efforts to explore areas that harbor the greatest biodiversity, such as aquatic environments, are needed to capture microorganisms that are rare and less abundant (*Schloss et al., 2016*). Traditionally, classification of microorganisms required that they be grown in pure culture. However, difficulties in mimicking specific environmental conditions that many marine bacteria require to grow in isolation have been a major impediment. The advancement of

Corresponding author
Cecilia Conaco,
cconaco@msi.upd.edu.ph

next-generation sequencing technologies has circumvented these challenges by allowing microbial community profiles to be directly obtained from environmental samples using the 16S rRNA gene as a barcode.

Understanding the biodiversity and the types of microbes that are present in a community is important as it provides information on ecosystem functioning (*Loreau et al., 2001*). The remarkable diversity of marine microbes can be attributed to their early evolution, rapid generation time, and the heterogeneity of the micro-environment (*Staley, 2006*). The heterogeneity of the ocean due to the presence of nutrient patches and microscale gradients (*Stocker, 2012*) results in different niches that can support diverse types of microbes (i.e., niche exclusion principle; *Kassen & Rainey, 2004*). In addition to these small-scale differences, large-scale spatial and temporal variations in the ocean also contribute to environmental heterogeneity and can sustain diversity (*Kassen & Rainey, 2004*). Other factors that may drive microbial diversity include dispersal, recombination, and coevolution through the processes of symbiosis and competition (*Kassen & Rainey, 2004*).

The archipelagic topology of the Philippines coupled with the range of geologic and oceanographic regimes (i.e., upwelling systems, anoxic basins, eutrophic coastal areas, and tectonically active regions, among others) provide diverse environments that may support biodiversity. Philippine waters host a diverse community of marine fishes, invertebrates, plants, and zooplankton (*Carpenter & Springer, 2005*; *Tittensor et al., 2010*), with which microorganisms may coevolve. Attributes such as habitat availability, heterogeneity, and sea surface temperature are said to be highly correlated with high species richness in the Philippines and nearby regions (*Sanciangco et al., 2013*). For these reasons, the waters surrounding the Philippine archipelago are likely to be areas of high microbial diversity.

The Benham Rise (also known as Philippine Rise) is an underwater plateau situated northeast of the Philippines, Western Pacific Ocean, where a convergence of waters occurs. Surface currents are mainly from North Pacific subtropical waters and Kuroshio recirculated waters, with inputs from the North Equatorial current, as well. These areas are vital to global ocean circulation and climate (*Gordon et al., 2014*; *Hu et al., 2015*). However, marine microbial taxa in the Western Pacific region are largely uncharted. These waters have not yet been explored by global efforts to sample the ocean microbiome, such as the Global Ocean Sampling (GOS) and Tara Expedition (*Parthasarathy, Hill & MacCallum, 2007*; *Sunagawa et al., 2015*). This study thus provides foundational data on the microbial diversity of the Benham Rise and reveals the association of microbial community structure with environmental factors.

## MATERIALS AND METHODS

### Physical and chemical measurements

Water samples were collected at different depths for physicochemical measurements focusing on the surface (SURF; 10 m), deep chlorophyll-*a* maximum (DCM; ∼90–210 m), and upper mesopelagic layer (UMP; 300 m) using Niskin bottles mounted on a rosette on board M/V DA BFAR on May 3–18, 2014 at Benham Rise (BR, Fig. 1). Nitrite, nitrate, silicate, and phosphate measurements were determined spectrophotometrically

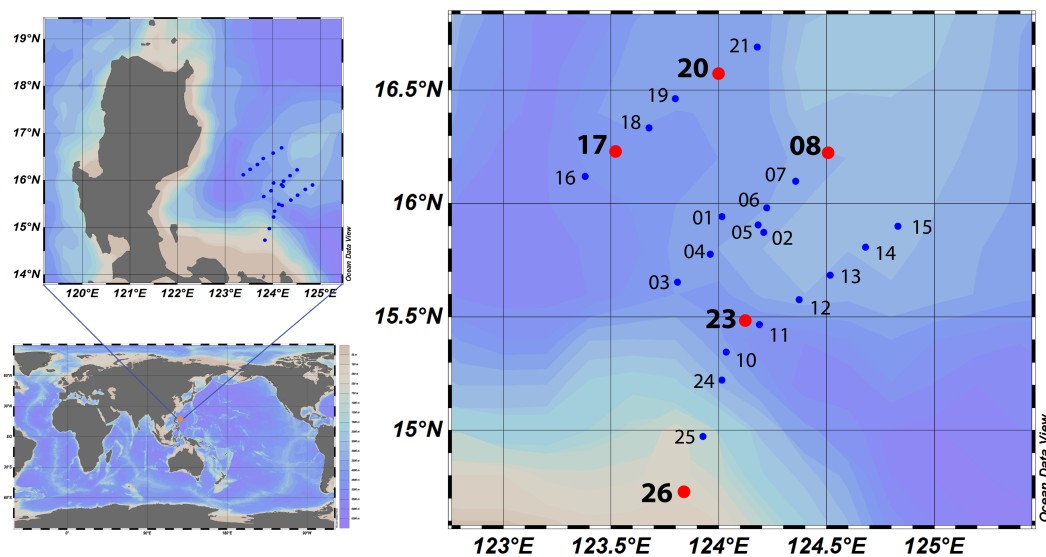

**Figure 1  Benham Rise, Philippines.** Benham Rise is an underwater plateau located northeast of the Philippines. Physicochemical parameters were measured at 24 stations (blue dots) while water samples for microbial community analysis were collected from various depths at five stations (red dots). Stations were plotted in Ocean Data View (Schlitzer, R., Ocean Data View, https://odv.awi.de, 2018).

following standard protocols (*Strickland & Parsons, 1972*) using a Skalar Sans$^{++}$ segmented flow analyzer D5000. Water samples for carbonate measurements were preserved using mercuric chloride and analyzed using a Kimoto total alkalinity titrator. Depth-profiles of conductivity, temperature, depth, dissolved oxygen (DO), and chlorophyll-*a* were determined using a Seabird$^{TM}$ SBE 19 plus attached to the rosette frame.

## On-board microbial sampling

Three depths were sampled for microbial analysis (SURF, DCM, and UMP) at five stations. Water samples were collected from Niskin samplers into autoclaved bottles. Approximately 1 L of water was pre-filtered through a sterile 20 μm mesh before filtering through a 0.2 μm polycarbonate filter. The size fraction collected captures both free-living and particle-attached microbes. The filter units were then placed in DNA lysis buffer (40 mM EDTA, 0.7 M sucrose, and 50 mM TrisCl) and frozen at −80 °C until extraction.

## DNA extraction

DNA was extracted using standard methods employing both enzymatic and bead beating homogenization (*De Boer et al., 2010*; *Huber, Butterfield & Baross, 2002*). Lysis and cell wall digestion were done with the addition of 40 μl of 50 mg/ml of lysozyme to thawed samples. Bead beating was carried out using ZR BashingBeads$^{TM}$ in a Precellys$^®$ homogenizer at 5,000 rpm for 3 × 60 s. Then, samples were incubated at 37 °C for 1 h. Further digestion was carried out by adding 50 μl of 20 mg/ml proteinase K and 100 μl of 20% SDS (sodium dodecyl sulfate) and incubating for 2 h at 55 °C. Organic extraction was done using phenol-chloroform-isoamyl alcohol (25:24:1) and subsequently with chloroform-isoamyl

alcohol (24:1). DNA was precipitated with an equal volume of isopropanol, washed with 70% ethanol, and resuspended in nuclease-free water.

## 16S rRNA amplification and sequencing

The V4 region of the 16S rRNA gene was amplified from the extracted genomic DNA using primers 515F (5′-GTGCCAGCMGCCGCGGTAA-3′) and 806R (5′-GGACTACHVGGGTWTCTAAT-3′), as previously described (*Caporaso et al., 2012*). Paired-end sequencing (250 bp) was performed on the Illumina MiSeq platform (Beijing Genomic Institute, Hong Kong) with an output of approximately 50,000–100,000 reads per sample. Raw sequencing reads are available on NCBI as BioProject number PRJNA386402.

## 16S rRNA sequence analysis

Assembly of paired-end reads into contigs and quality filtering were implemented following the mothur MiSeq pipeline (v1.35.1) (*Kozich et al., 2013*; *Schloss et al., 2009*). Assembled contigs were aligned to the SILVA version 132 database (*Quast et al., 2013*). Reads were checked for chimeric sequences using the Uchime algorithm (*Edgar et al., 2011*). Sequences were clustered into operational taxonomic units (OTU) at 97% similarity cutoff. OTUs were taxonomically classified using the SILVA version 132 database. Diversity, richness, and community comparisons were calculated using mothur. Statistical tests including (1) Parsimony test, (2) weighted and unweighted Unifrac, (3) AMOVA, (4) HOMOVA, and (5) $\int$-LIBSHUFF were implemented in mothur to test whether microbial communities across depths have similar structure (*Schloss, 2008*). LEfSe (*Segata et al., 2011*) and indicator analysis (*Dufrêne & Legendre, 1997*) were implemented in mothur to identify overrepresented OTUs.

## Integrating physicochemical and 16S rRNA sequence data

Canonical correspondence analysis (*Ter Braak, 1986*) was implemented in XLSTAT to explain the variation in microbial communities, specifically to relate species abundance to environmental condition. The relative abundance of major taxa were square-root transformed for normalization while variables, such as nitrite + nitrate, phosphate, silicate, and chlorophyll-a concentration, were log $(x + 1)$ transformed (*Ramette, 2007*). Samples with undetectable concentrations of phosphate and nitrite + nitrate were set to zero before transformation. Variables such as temperature, salinity, DO, pH, and turbidity were not data transformed.

# RESULTS

## Oceanography of Benham Rise

Benham Rise, bounded by the coordinates 119°30′E to 132°00′E longitude and 12°10′N to 20°30′N latitude, is an underwater inactive volcano (*United Nations (UN), 2009*; *Savov et al., 2005*). It is a plateau, which stands 3,500 m and 500 m above the surrounding seafloor at its crest and northern-eastern margins, respectively. Oceanographic measurements and bacterial sampling were done around Benham Bank (∼50 m deep), the shallowest portion of Benham Rise. Twenty-four stations were occupied around Benham Bank for

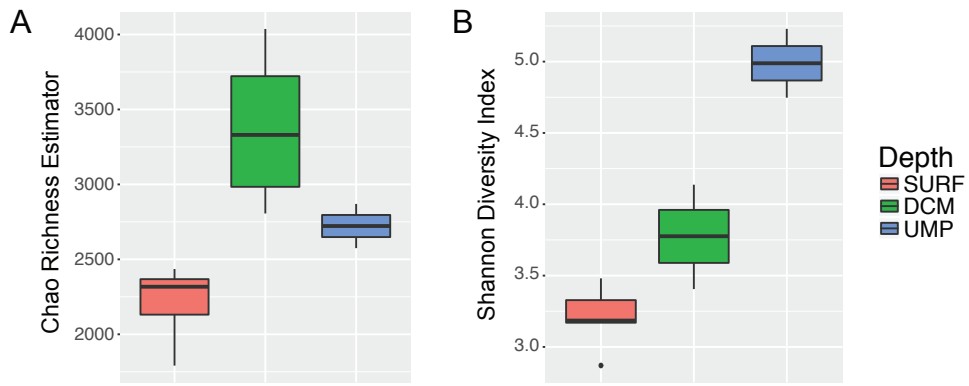

**Figure 2** **Depth-related patterns in microbial richness and diversity.** (A) Species richness, as estimated using the Chao index, and (B) diversity, based on the Shannon index, increase with depth.

physicochemical measurements, five stations were also sampled for microbial community analysis (Fig. 1 and Table S1). In total, five surface (SURF), four deep-chlorophyll maximum (DCM), and two upper mesopelagic (UMP) samples were obtained. Stratification of physical and chemical properties was evident in the water column. A strong gradient in temperature and salinity was found at the upper 20–40 m and at 200–400 m (Figs. S1A, S1B and Table S2). Deep chlorophyll maximum was detected at around 110 to 150 m (Fig. S1C) while oxygen minima were detected at 200 m and at 750 m (Fig. S1D). Nutrient concentration increased with depth as expected for a stratified, oligotrophic ocean, while pH decreased with depth (Fig. S2 and Table S3).

## Microbial community composition and structure

A total of 837,124 reads were pooled from 11 samples. Removal of contigs with ambiguous bases and reads with length >275 resulted in the rejection of 25% of the initial reads. Elimination of chimaeras and removal of lineages corresponding to chloroplast, eukaryotes, mitochondria, and unknown sequences, resulted in the removal of 21.7% and 1.4% of the contigs, respectively. After quality filtering steps, 483,773 contigs corresponding to 37,659 unique sequences remained. These sequences were classified into 10,599 OTUs (4,887 OTUs without singletons). The non-plateauing rarefaction curves suggest that Benham Rise waters host a much more diverse prokaryotic community than reported here (Fig. S3). There is also an increasing trend in indices of species richness and diversity with increasing depth (Fig. 2).

Samples from the same depth taken at different stations exhibited more similar community composition than samples from different depths taken at the same stations (Fig. 3A). Only 441 OTUs were common to all depths (Fig. 3B). Two thousand, nine hundred eighty OTUs were specific to the surface samples, 3,791 to the DCM, and 1,837 to the UMP. More OTUs overlapped between surface and DCM (789) and between DCM and UMP (692) compared to surface and UMP (69). Significant differences in microbial community structure across depths were supported by weighted UniFrac and AMOVA ($p < 0.05$) but not by unweighted UniFrac ($p > 0.05$), and HOMOVA ($p > 0.05$) (Table S4).
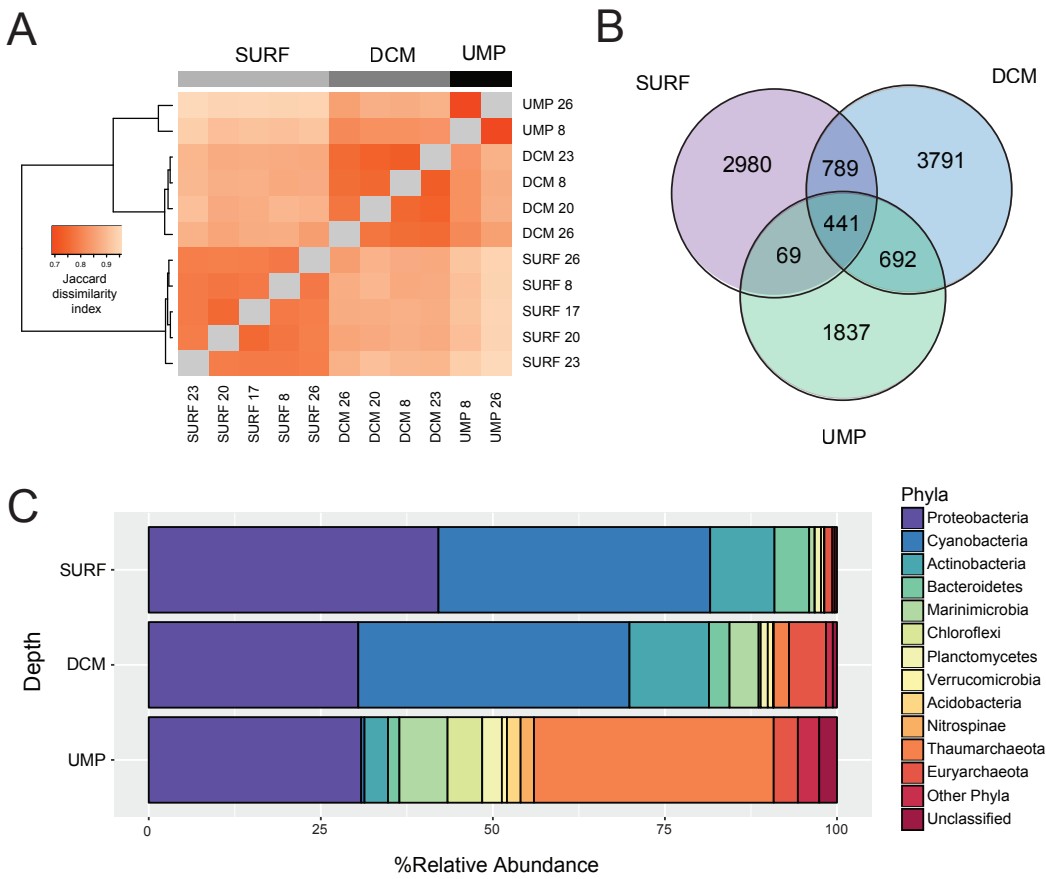

**Figure 3** **Microbial community composition and structure in the Benham Rise water column.** (A) Samples from the same depth taken at different stations exhibit more similar community composition (SURF, surface; DCM, deep chlorophyll maximum; UMP, upper mesopelagic). The heatmap represents the Jaccard dissimilarity index between samples with dark orange signifying high similarity. (B) The number of OTUs that are shared among the three depths or that are unique to specific depths. (C) The relative abundance of microbial phyla at three depths. For simplicity, less abundant groups were classified under "other phyla."

Only the surface and DCM communities were significantly different under the Parsimony test ($p < 0.05$) and $\int$-LIBSHUFF ($p < 0.025$ for two $p$-values) using Jaccard distances as input (Table S4).

The 10 most dominant bacterial phyla detected in the Benham Rise water column were Proteobacteria, Cyanobacteria, Actinobacteria, Bacteroidetes, Marinimicrobia, Chloroflexi, Planctomycetes, Verrucomicrobia, Acidobacteria, and Nitrospinae, while Thaumarchaeota and Euryarchaeota were the major archaeal phyla (Fig. 3C). Cyanobacteria constituted a major proportion of surface and DCM sequences, with 16%–43% of sequences belonging to *Prochlorococcus* and 0.1%–9.31% to *Synechococcus*. Other ecologically relevant marine taxa that were detected at all sampled depths were SAR 11 (0.05%–1.45%), SAR86 (0.9%–8%), SAR116 (0.01%–4%), SAR406 (Marine Group A) (0.5%–8.7%), SAR324 (Marine Group B) (0.05%–4.7%), and *Alteromonas* (0.15%–16%). SAR202 (4%) and archaeal

groups, including Nitrosopumilaceae (21%–48%), a family within Marine Group I (*Konneke et al., 2005*), as well as Marine Group II (2–3%), Marine Group III (1%), and Candidatus *Nitrosopelagicus* (4%–7%), were abundant in the UMP layer. Some OTUs were significantly overrepresented at different depths, notably Otu00007 (SAR86) and Otu00015 (SAR116) in the surface, Otu00009 (Marine Group II) in the DCM, and Otu00005 (Nitrosopumilaceae) in the UMP (Table S5). Indicator analysis revealed OTUs that are responsible for differences in groupings of samples, including Otu00310 (*Spirochaeta*) and Otu00702 (*Nitrospina*) for UMP (Table S5).

## Physicochemical parameters affecting microbial abundance

CCA showed association of major phyla with physicochemical properties of the water column (Fig. 4). Samples from the same depth grouped together in CCA ordination, which agreed with the depth-related distribution patterns revealed by Jaccard index, weighted UniFrac, and AMOVA. Although analysis was based on a limited number of samples, taxa-environment relationships that have previously been established were observed. For instance, oligotrophs like SAR11 clade of Alphaproteobacteria, SAR86, and *Prochlorococcus* were associated with low nutrient surface water (Fig. 4). Moreover, cyanobacteria were shown to be associated with the surface and DCM samples. Thaumarcheaota, dominated by Nitrosopumilaceae, were strongly associated with high levels of nitrite and nitrate (Fig. 4).

## DISCUSSION

In this study we generated baseline data on the taxonomic diversity of bacteria and some archaea in an exploratory survey of Benham Rise. A total of 10,599 OTUs were recovered by sequencing of the 16S rRNA V4 region. By comparison, the Tara Oceans Expedition uncovered 35,000 prokaryotic OTUs in the euphotic zone by means of whole genome shotgun sequencing (*Sunagawa et al., 2015*) while the GOS Expedition recovered 811 distinct ribotypes from clustering 4,125 full and partial length 16S at 97% similarity sampling mostly from the surface layer (*Rusch et al., 2007*). It should be noted that the 515F and 806R V4 primers used in the present study are predicted to detect only about 86.8% of Bacteria and 52.9% of Archaea based on in silico evaluation using SILVA TestPrime (*Klindworth et al., 2013*). These primers have also been shown to underrepresent the SAR11 clade and some Thaumarchaeota, while overestimating Gammaproteobacteria (*Apprill et al., 2015*; *Parada, Needham & Fuhrman, 2016*). In addition, taxa under the candidate phyla radiation (CPR) will evade detection using this V4 primer set (*Brown et al., 2015*).

The species richness and diversity of the microbial community in surface waters of the Benham Rise is within the range of richness and diversity in adjacent surface waters of the Indian Ocean-South China Sea (*Zheng, Dai & Huang, 2016*). Furthermore, the increase in microbial community richness and diversity with depth is similar to findings reported in the global Tara Oceans survey. This trend may be explained by an increase in the variety of ecological niches provided by marine snow microenvironments (*Stocker, 2012*; *Sunagawa et al., 2015*), as well as slow growth and higher motility at the mesopelagic layer, which decreases predation and viral lysis (*Pernthaler, 2005*; *Sunagawa et al., 2015*).

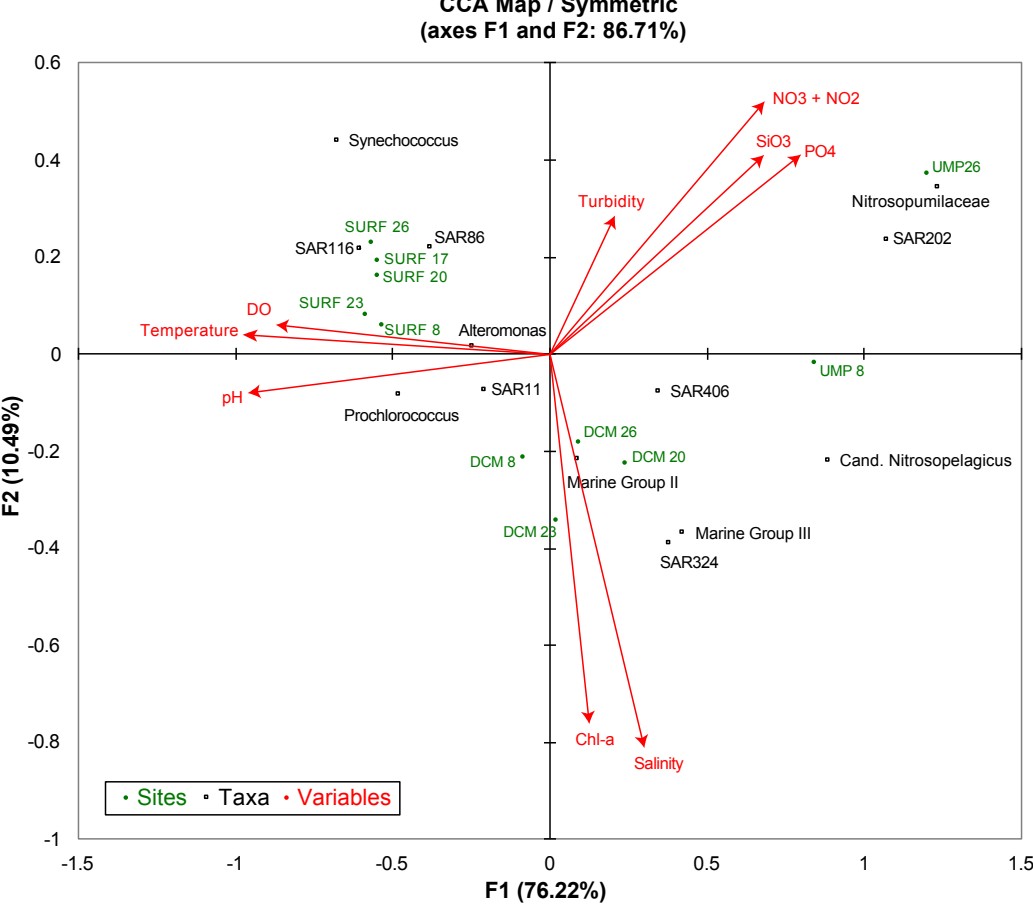

**Figure 4  Canonical correspondence analysis (CCA) ordination plot.** The relationship between sampling sites and depths (green), microbial taxa (black), and specific environmental parameters (red arrows) are shown. The combination of the environmental variables explained 86.71% of the total variance in the relative abundance of selected major taxa in Benham Rise.

## Microbial community composition

The general composition of the microbial community in Benham Rise is similar to that described from other marine environments by studies such as the GOS (*Biers, Sun & Howard, 2009*; *Yooseph et al., 2007*), Tara Oceans Expedition (*Sunagawa et al., 2015*), and others (*DeLong et al., 2006*; *Yilmaz et al., 2015*). There is a high incidence of sequences belonging to Alphaproteobacteria, Gammaproteobacteria, and Cyanobacteria (*Prochlorococcus* and *Synechococcus*) in the surface and DCM waters, and a higher frequency of archaea in the mesopelagic layer. As expected, a large portion of 16S rRNA amplicons belong to the genera *Synechococcus* and *Prochlorococcus*, which comprise the most abundant cyanobacteria in the global ocean and are central to primary productivity and carbon cycling (*Biller et al., 2015*; *Scanlan & West, 2002*). Another taxon that is similarly abundant is *Alteromonas* sp., which is hypothesized to be crucial for dissolved organic carbon (DOC) consumption in the upper ocean layer (*Pedler, Aluwihare & Azam, 2014*).

SAR324 (under class Deltaproteobacteria) and SAR406 (a bacterial phylum also known as Marine Group A) were observed to be relatively high in DCM and UMP samples, consistent with their enrichment in deep waters in both the Atlantic (*Agogue et al., 2011*) and the Pacific (*DeLong et al., 2006*; *Pham et al., 2008*). Similarly, SAR202 clade is generally associated with the aphotic zone (*Morris et al., 2004*). The underrepresentation of SAR11 (also known as order *Pelagibacterales* under Alphaproteobacteria) is also notable and is presumably a result of the bias in the V4 primers used in this study. In the BR dataset, only 0.05%–1.45% of reads belonged to SAR11 whereas 30.9% of reads in the GOS dataset belonged to this group (*Biers, Sun & Howard, 2009*). It is estimated that 25%–50% of the microbial community in the ocean is dominated by SAR11, half of which resides in the euphotic zone (*Giovannoni, 2017*; *Morris et al., 2002*). *Pelagibacter*, a member of the SAR11 clade, was recently shown to assimilate dimethylsulfoniopropionate (DMSP) producing dimethyl sulfide (DMS), an important compound for climate regulation (*Sun et al., 2016*). Another DMS-producer, SAR116 (under class Alphaproteobacteria), which has been shown to be abundant in the northwest Pacific Ocean (*Choi et al., 2015*), was also detected in the BR dataset. The abundance of archaea in the Benham Rise UMP corroborates the finding that archaea dominate the mesopelagic layer of the Pacific Ocean (*Karner, DeLong & Karl, 2001*). Of the archaeal taxa, Marine Group I and Marine Group II are the cosmopolitan groups in the ocean (*Massana, DeLong & Pedros-Alio, 2000*). Thaumarchaeota dominates the UMP while Euryarchaeota are less abundant in the deep sea (*Yilmaz et al., 2015*).

## Depth-related stratification of microbial community

Different statistical tools suggest depth-related stratification of the BR microbial community. The influence of vertical stratification on microbes has also been observed by other studies (*DeLong et al., 2006*; *Hewson et al., 2006*; *Treusch et al., 2009*). This pattern of microbial community distribution is likely attributed to the differences in the physicochemical properties of the water column that act as a dispersal barrier and lead to the formation of specific microbial communities in different water masses (*Agogue et al., 2011*). The greater number of depth-specific OTUs compared to shared OTUs suggests the presence of many specialist taxa (Fig. 3B). Although they have a narrow utilization range, specialist taxa have high peak performance and high growth rates (*Mariadassou, Pichon & Ebert, 2015*). These features, along with physicochemical barriers that limit competition and invasion, favor resident specialist taxa, which are generally more dominant in diverse habitats (*Mariadassou, Pichon & Ebert, 2015*).

On the other hand, the ubiquity of some bacterial clades is explained by their metabolic versatility. For instance, the ubiquitous SAR324 clade displays a wide range of metabolic capabilities, including lithotrophy, heterotrophy, and alkane oxidation (*Sheik, Jain & Dick, 2014*). Genomic evidence for a SAR116 representative shows features of a metabolic generalist (*Oh et al., 2010*). SAR406 (also known as Marine Group A or Marinimicrobia), which has a role in the sulfur cycle, is abundant in the upper ocean but is also found in oxygen minimum zones (OMZ) and anoxic basins, suggesting metabolic versatility (*Wright et al., 2014*).

Prokaryotes that are overrepresented at particular depths may have the ability to adapt to specific conditions encountered at those depths (Table S5). For example, species of *Spirochaeta* are known facultative anaerobes that can survive low oxygen levels in the UMP (*Breznak & Warnecke, 2008*). In addition, *Nitrospina* species that are overrepresented in the UMP where nitrites are abundant are known nitrite oxidizers (*Lucker et al., 2013*; *Spieck et al., 2014*). It is important to note, however, that 16S rRNA surveys are almost always incomplete. Depth specificity does not always equate to the absence of that particular taxa at other depths. Differences in sampling time and small-scale geographic variability might also influence the detection of certain taxa.

Various factors can affect microbial community structure, such as physical and chemical conditions, dispersal, predation, grazing, viral lysis, resource availability, and environmental variability, among others (*Agogue et al., 2011*; *Follows & Dutkiewicz, 2011*; *Pedros-Alio, 2006*). Here we examined the association of physicochemical properties of the water column with major microbial phyla. CCA was able to recapitulate previously known phyla-environment associations. Clades that are adapted to oligotrophic water, such as SAR11, SAR86, and *Prochlorococcus*, clustered within the surface samples and were negatively correlated with nutrients (Fig. 4). The small size of *Prochlorococcus* cells is an adaptation to a low nutrient environment (*Partensky, Hess & Vaulot, 1999*). SAR11, on the other hand, has a streamlined genome that allows it to thrive in low nutrient environments (*Giovannoni, Trash & Temperton, 2014*; *Giovannoni et al., 2005*). As expected for oligotrophic prokaryotes (*Mayali, Palenik & Burton, 2010*), SAR11 and SAR116 were also found to be negatively correlated with chlorophyll-a. On the other hand, the distinct association of the archaeal family, Nitrosopumilaceae, with high levels of nitrite and nitrate is consistent with their known ammonia-oxidizing capabilities (*Konneke et al., 2005*).

## CONCLUSIONS

The diverse oceanographic regimes, rich geologic history, and biogeographic novelty of the Philippine archipelago offer an excellent opportunity to examine fundamental questions in marine microbial ecology and biogeography. The work presented here presents an initial glimpse into marine microbial diversity in Philippine waters. By means of high-throughput sequencing of the V4 region of the 16S rRNA gene, we reveal that the microbial composition, richness, and diversity of waters in the Benham Rise are similar to other tropical and subtropical open ocean regions. Benham Rise exhibits vertical zonation of marine microbes with a greater abundance of specialist taxa at different depths and increased biodiversity in the mesopelagic layer. The presence of specific phyla could be correlated with physicochemical properties of the water column. Finally, it is important to note the limitations of this and other community analyses that rely on amplification of 16S rRNA. Specifically, primer bias in the detection of certain phyla, within-ribotype diversity resolution, and microbial functions cannot necessarily be inferred using a targeted, single-gene approach. Further studies with more extensive sampling and that make use of other broad-spectrum primers, whole metagenome sequencing, or single cell genomics will shed more light on archaeal and bacterial diversity, as well as the structure and function of microbes in this region of the ocean.

## ACKNOWLEDGEMENTS

The authors would like to thank the officers and crew of the MV DA-BFAR of the Bureau of Fisheries and Aquatic Resources of the Philippines. The authors would also like to acknowledge Grieg F. Steward for his valuable comments and suggestions.

### Funding

This study was funded by a UP Marine Science Institute in-house grant to Cecilia Conaco. The research cruise was funded by the Department of Science and Technology and also supported by the Department of Agriculture—Bureau of Fisheries and Aquatic Resources of the Philippines, specifically through the use of the MV DA-BFAR vessel. The funders had no role in study design, data collection and analysis, decision to publish, or preparation of the manuscript.

### Grant Disclosures

The following grant information was disclosed by the authors:
UP Marine Science Institute.
Department of Science and Technology.
Department of Agriculture—Bureau of Fisheries and Aquatic Resources of the Philippines.

### Competing Interests

The authors declare there are no competing interests.

### Author Contributions

- Andrian P. Gajigan conceived and designed the experiments, performed the experiments, analyzed the data, prepared figures and/or tables, authored or reviewed drafts of the paper, approved the final draft.
- Aletta T. Yñiguez, Cesar L. Villanoy, Maria Lourdes San Diego-McGlone and Gil S. Jacinto contributed reagents/materials/analysis tools, authored or reviewed drafts of the paper, approved the final draft.
- Cecilia Conaco conceived and designed the experiments, analyzed the data, prepared figures and/or tables, authored or reviewed drafts of the paper, approved the final draft.

### DNA Deposition

The following information was supplied regarding the deposition of DNA sequences:
Raw sequencing reads are available on NCBI as BioProject number PRJNA386402.

### Data Availability

The raw data are uploaded in the Supplemental Files.

### Supplemental Information

Supplemental information for this article can be found online at http://dx.doi.org/10.7717/peerj.4781#supplemental-information.

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
