# Peer review of "Diversity and community structure of marine microbes around the Benham Rise underwater plateau, northeastern Philippines"

_PeerJ, doi:10.7717/peerj.4781_

## Round 0.1 · original submission · Major Revisions

Please take a close look at the reviewer's comments. I agree the manuscript suffers based on current organization. Too much emphasis is placed on the analysis pipeline, especially with so few samples. This should be toned down. You may want to consider updating your analysis with the SILVA 132 release as well.

·

Basic reporting

The article is well-written with standard structure and sufficient references.

Experimental design

This article fills a gap by exploring the diversity of microbes in the marine environment near the Philippines. The methods generally followed standards in the field.

I do have some comments:

L142: says SILVA release 102 was used. This database is from 2010 and is only 25% the size of the current release, SILVA 132, with poorer curated alignments. Mismatches and poor alignments to the reference may partially account for the high number of OTUs found.

The mothur-RDP sections and inter-method comparisons could (and I think probably should) be removed because the comparison between pipelines is not completed on enough samples to be broadly useful beyond this publication. The obvious lack of representation of the expected community also argues against the usefulness of the mothur-RDP dataset.

Validity of the findings

The main issue, as mentioned by previous reviewers, is the limited number of samples (N=11 in 3 depth bins). This is useful for a first-pass pilot study but rigorous statistical treatment really isn't feasible with so few samples.

L224: two of these are obligate anaerobes -- how did they get into surface waters?

Figure 3. This comparison between studies is invalid because of the different methods used. The authors might still discuss the general trend of higher diversity with depth, but the surface-only comparisons with other seas aren't meaningful and should be removed. People have found that just about everything affects these diversity measurements, so unless you use identical collection, filtering, DNA extraction methods, primers, PCR conditions, Illumina library kits and methods, demultiplexing, software versions, and database versions, you shouldn't be comparing these numbers (that's why EMP, Tara, OSD, etc. have standardized all of these methods). And *especially* not between metagenomic and amplicon sequencing methods. It's not telling you about the communities, only the methods, and even then it's not telling you very much.

Additional comments

I commend the authors on the extensive metadata they've provided with the manuscript. It is a valuable dataset and I look forward to its eventual publication.

Reviewer 2 ·

Basic reporting

The manuscript meets the criteria required for basic reporting.

Experimental design

The manuscript is a unique primary research study. There are some issues to resolve regarding experimental design prior to publication. More specifically, the question needs to be better defined given the data presented. The introduction justifies the need to resolve microbial diversity in this part of the ocean. However, much of the paper and data, figures and supplement are focused more on comparing 16S rDNA community characterization pipelines. There are several problems with this (see general comments for the author).

The manuscript needs to be reorganized. It's not clear to me why results from the Mothur MiSeq pipeline were discussed in great detail, and before results from the SILVA pipeline, especially given the clear indication that SILVA produced much better results.

Many of the figures are not discussed in sufficient detail. Very little information is provided in the results section to help the reader interpret the figures and many of the supplemental figures are not needed. Data in many of the figures and tables are introduced only briefly (a single sentence) and/or presented at different places in the results section. This makes it very difficult to follow. Specific example (line 203), the first time Figure 1B is introduced there is no mention of the different 16S rDNA analysis pipelines. Specific example, Figures S4, S5 and S6 are introduced in only briefly (line 212-217).

Validity of the findings

Much of the supplement is not needed. It is either presented sufficiently in the text or not discussed in enough detail in the results. Some examples are listed below:

The data in Fig. S2 would be more useful to the community if they were provided in a table. The plots are OK but not really needed since there is very little difference and the results are expected for this type of system. Data in a table could be used by others for modeling or other analyses that need the numbers. Some discussion about the data in the results could help. For example, what is the range for some of the important variables plotted? Key features, depths indicating changes in water masses, etc.?

The word contigs is used. Are these contigs (overlapping sequence reads) or are they individual sequence reads?

Some discussion to help explain Figure 1B line 203 is needed. What are some of the observed similarities with analysis method and sample type? What are some of the differences? This sentence gives the impression that there should only be one bar in the graph for the relative abundance of these groups. What is the significance of the different bar graphs for these groups? SILVA appears to do a much better job. Why show differences in the analysis methods? This is mentioned later (line 256) for the SILVA analysis, but by the time I got that far I was already confused about the purpose of the figure.

Additional comments

Reorganize the manuscript using the best data/analysis about diversity in this part of the ocean. Differences in analysis methods are secondary, if included at all. If they are included, consider presenting the more accurate data (SILVA) first and discussing the relevance. Present a methods comparison last.

External reviews were received for this submission. These reviews were used by the Editor when they made their decision, and can be downloaded below.

---

## Round 0.2 · accepted · Accept

Authors have successfully incorporated reviewer's suggested edits. Manuscript reads much better now.

#